# A Double Fourier-Transform Imaging Algorithm for a 24 GHz FMCW Short-Range Radar

**DOI:** 10.3390/s23084119

**Published:** 2023-04-19

**Authors:** Renato Cicchetti, Stefano Pisa, Emanuele Piuzzi, Orlandino Testa

**Affiliations:** Department of Information Engineering, Electronics and Telecommunications, Sapienza University of Rome, 00184 Rome, Italy; renato.cicchetti@uniroma1.it (R.C.); emanuele.piuzzi@uniroma1.it (E.P.);

**Keywords:** delay and sum (DAS), double Fourier transform (2D-FT), multiple signal classification (MUSIC), short range radar imaging system, serial patch array antenna

## Abstract

A frequency-modulated continuous-wave radar for short-range target imaging, assembling a transceiver, a PLL, an SP4T switch, and a serial patch antenna array, was realized. A new algorithm based on a double Fourier transform (2D-FT) was developed and compared with the delay and sum (DAS) and multiple signal classification (MUSIC) algorithms proposed in the literature for target detection. The three reconstruction algorithms were applied to simulated canonical cases evidencing radar resolutions close to the theoretical ones. The proposed 2D-FT algorithm exhibits an angle of view greater than 25° and is five times faster than DAS and 20 times faster than the MUSIC one. The realized radar shows a range resolution of 55 cm and an angular resolution of 14° and is able to correctly identify the positions of single and multiple targets in realistic scenarios, with errors lower than 20 cm.

## 1. Introduction

In the last years, radar systems have been widely used in defense, automotive, navigation, traffic control, and environmental monitoring sectors, as well as medicine and sports [1]. The main advantages of radars, with respect to ultrasound (sonar) and optical (LIDAR) systems, are their relative insensitivity to weather conditions (rain, snow, dust, etc.), the possibility to accurately identify target movements, and the ability to penetrate non-metallic materials. Various types of radars, such as Doppler [2,3], ultra-wideband [4,5], and frequency-modulated continuous-wave (FMCW) [6,7] radars, were proposed in the literature.

An FMCW radar for the berthing of large ships in inclement weather with poor visibility was proposed in [8]. The radar operates at a central frequency of *f*_0_ = 9.25 GHz, with a bandwidth of *B* = 100 MHz and employs 4 transmitting (TX) and 16 receiving (RX) antennas, synthesizing, in this way, a uniform linear array of 64 virtual antennas. Image reconstruction was achieved by employing the beam-steering technique for different identification distances. A multi-beam and multi-range FMCW radar (*f*_0_ = 24 GHz, *B* = 100 MHz) was presented in [9]. The proposed radar can be used for cruise control and stop-and-go systems. A two-channel TX and four-channel RX structure was designed and connected to series-fed patch antennas. In order to extract angular information with high accuracy, the subspace-based digital beam forming (DBF) with a multiple signal classification (MUSIC) algorithm was used. In [10], a surveillance FMCW radar (*f*_0_ = 24 GHz, *B* = 2 GHz) to monitor car and pedestrian traffic, as well as intruders, was designed. The developed radar system consists of one transceiver with two selectable TX antennas, eight parallel RX antennas, and a back-end module for data logging and control. In the target detection algorithms, the ranges and relative velocities of multiple targets were obtained by means of 2D Range-Doppler-FFT processing, while the angular information was determined by digital beam forming based on the received signals.

Further applications of FMCW radars concerned the monitoring of physical activity of subjects [11,12], the classification of hand movements using convolutional neural networks [13,14], the localization and monitoring of vital signatures of subjects [15], and the recognition of the smoothness of railway tracks [16].

Various types of FMCW transceivers are available on the market that operate in the 24 GHz industrial, scientific, and medical (ISM) band [17,18,19,20,21,22]. Distance2Go is a 24 GHz radar based on the BGT24MTR11 transceiver chipset from Infineon [17]. The radar board, equipped with one TX and one RX antenna, can measure the distance, speed, and direction of movement. These features make the radar suitable for various applications, such as motion detection, presence sensing, proximity sensing, etc. [18,19]. The EV-TINYRAD24G from Analog-Devices is a 24 GHz radar evaluation module that allows for the implementation and testing of radar-sensing applications [20]. This radar has two TX and four RX integrated antenna array in a multiple-input/multiple-output (MIMO) radar scheme. The combined antennas allow the TinyRad module to detect the distance, speed, and angular position of multiple targets simultaneously. The IMST 24 GHz FMCW radar Development Kit DK-sR-1200e is provided with high level human-tracker and range-finder algorithms. The radar has one TX and two RX channels equipped with integrated patch antennas [21] and was used to monitor the evolution of lake ice thickness and snow mass [22].

In the mentioned papers, very complex and integrated radar realizations were presented. Moreover, only one reconstruction algorithm was used, and no comparison between different techniques was reported.

In this work a simple and modular FMCW radar for the control of subjects on the sidewalks of train and subway stations and for monitoring elderly subjects in their home environment was designed and realized. Since, for the abovementioned applications, fast acquisition times are required, a system with one TX and four RX antennas was chosen. A new, fast, and simple reconstruction algorithm based on double-fast Fourier transform (2D-FT) was proposed and compared with already available delay and sum and MUSIC algorithms [23]. To evaluate the overall performance of the designed radar system, both the simulated and experimental results are presented.

## 2. 2D–FT Reconstruction Algorithm

To generate a two-dimensional image of a scenario by using radar data or numerical simulations, suitable reconstruction algorithms must be used. Five different reconstruction algorithms, used to process data acquired with a frequency step radar (SFR), were presented in [23]. In the radar proposed in [23], the scattering parameters (S21) between M selectable pairs of transmitting (TX) and receiving (RX) antennas (MIMO radar) were measured for a given number (N) of frequencies.

Since FMCW radars use IQ demodulators, the two IF outputs generated by mixing the received and transmitted signals are proportional to the real and the imaginary part of the transmission scattering parameter, which, in turn, are proportional to the target radar cross-section. In conclusion, the reconstruction algorithms presented in [23], with reference to a frequency step radar, can also be applied to the present case concerning an FMCW radar. Two different algorithms, the delay and sum (DAS) and the multiple signal classification (MUSIC) are used in the following. Moreover, since the fast Fourier (FT) algorithm described in [23] does not work properly with few antennas (M = 4 in this case), a new double Fourier transform (2D-FT) intensity function is introduced to overcome this drawback.

The normalized 2D-FT intensity function can be derived by the DAS intensity function given by the following:(1)$$I_{DAS}\left(x,y\right)=\frac{F_{DAS}\left(x,y\right)}{F_{DASMax}}$$
(2)$$F_{DAS}\left(x,y\right)=\left|\int_{f_{0}-\frac{B}{2}}^{f_{0}+\frac{B}{2}}\int_{-\frac{L}{2}}^{+\frac{L}{2}}\Gamma \left(\xi ,f\right)e^{+j\frac{4\pi f}{c}\sqrt{{\left(x-\xi \right)}^{2}+y^{2}}}d\xi df\right|$$
where $F_{DASMax}$ is the maximum value returned by $F_{DAS}\left(x,y\right)$ in the Region of Interest (ROI), $f_{0}$ is the central frequency of the operating band B, and L is the length of the array centered in the origin of the Cartesian reference system (see Figure 1).

A second-order Taylor expansion of the phase term appearing in the integrand function in (2) is introduced to reduce its numerical computational burden. By doing so, as well as by employing the approximation reported in Appendix A, we obtain the following:(3)$$F_{DAS}\left(x,y\right)≅\left|\int_{f_{0}-\frac{B}{2}}^{f_{0}+\frac{B}{2}}\int_{-\frac{L}{2}}^{+\frac{L}{2}}\Gamma \left(\xi ,f\right)e^{+j\frac{4\pi f}{c}\sqrt{x^{2}+y^{2}}-j\frac{4\pi f_{0}}{c}\frac{x\xi }{\sqrt{x^{2}+y^{2}}}}d\xi df\right|=\;={\left|\int_{f_{0}-\frac{B}{2}}^{f_{0}+\frac{B}{2}}\int_{-\frac{L}{2}}^{+\frac{L}{2}}\Gamma \left(\xi ,f\right)e^{+j2\pi f\tau +j2\pi \gamma \xi }d\xi df\right|}_{\begin{array}{l} \gamma =-\frac{2f_{0}}{c}\frac{x}{\sqrt{x^{2}+y^{2}}}\\ \tau =\frac{2}{c}\sqrt{x^{2}+y^{2}} \end{array}}=\left|F_{2D-FT}\left(x,y\right)\right|$$

Equation (3) allows us to define the intensity function, $I_{2D-FT}\left(x,y\right)$, based on the 2D inverse Fourier transform through the following analytical link between the variables (*γ*, *τ*) and the observation point (*x*, *y*):(4)$$\begin{array}{l} \gamma =-\frac{2f_{0}}{c}\frac{x}{\sqrt{x^{2}+y^{2}}}\\ \tau =\frac{2}{c}\sqrt{x^{2}+y^{2}} \end{array}$$
thus obtaining
(5)$$I_{2D-FT}\left(x,y\right)≅\frac{\left|F_{2D-FT}\left(x,y\right)\right|}{\left|F_{2D-FTMax}\right|}$$
where $\left|F_{2D-FTMax}\right|$ is the maximum magnitude value returned by $F_{2D-FT}\left(x,y\right)$ in the ROI.

Figure 2 shows the mapping of the ROI in the corresponding region expressed by means of the γ and τ variables, where
(6)$$\gamma_{min}=-\frac{2f_{0}}{c}\frac{x_{max}}{\sqrt{x_{max}^{2}+y_{min}^{2}}}$$
(7)$$\gamma_{max}=-\frac{2f_{0}}{c}\frac{x_{min}}{\sqrt{x_{min}^{2}+y_{min}^{2}}}$$
(8)$$\tau_{min}=\frac{2y_{min}}{c}$$
(9)$$\tau_{max}=\frac{2d_{max}}{c}$$
and $d_{max}$ is the maximum distance of the ROI points from the origin of the Cartesian reference system.

The mapping from the γ and τ domain to the spatial variables x and y is expressed as follows:(10)$$\begin{array}{l} x=-\frac{c\tau \gamma }{4}\lambda_{0}\\ y=\frac{c\tau }{2}\sqrt{1-{\left(\frac{\lambda_{0}\gamma }{2}\right)}^{2}} \end{array}$$

If the following condition is met,
(11)$$\frac{\lambda_{0}\left|\gamma \right|}{2}\ll 1$$
which, according to (4), implies
(12)$$\left\{\begin{array}{c} \frac{\left|x_{min}\right|}{y_{min}}\ll 1\\ \frac{\left|x_{max}\right|}{y_{min}}\ll 1 \end{array}\right.$$
then (10) can be adequately approximated as follows:(13)$$\begin{array}{l} x=-\frac{\gamma \lambda_{0}}{2}y\\ y=\frac{c\tau }{2} \end{array}$$

Note that (12) corresponds to a ROI located far from the Radar system array, i.e., a situation of practical interest.

The numerical implementation of the proposed 2D-FT reconstruction algorithm is reported in Appendix B.

## 3. Radar DESIGN

Figure 3a shows the schematic of the designed modular radar. The EVAL-BGT24MTR11 board by Infineon was used as the transceiver [24]. The BGT24MTR11 chip contains a VCO operating between 24 and 25.5 GHz, power and low-noise amplifiers, and an I-Q mixer (see Figure 3b). The scaled by 16 board output frequency (Q1) is sent to the VCO input of the EVAL ADF4159 board by Analog Devices used as PLL circuit (see Figure 3c) [25]. The charge pump output (CP) of the board is filtered by the loop filter and sent to the VCO control input (FINE/COARSE). By properly setting the registers of the ADF4159, the BGT24MTR11 board generates a chirp signal in the ISM band between 24 and 24.25 GHz, with a fixed time duration. One of the two balanced outputs of the transceiver is sent to the transmitting antenna (TX), while the other (TXX) is terminated on a 50 Ω load. The RF output power of the board is about 11 dBm, with a power consumption of 500 mW in continuous operating mode. The low-noise amplifier noise figure is 12 dB, and the mixer conversion gain is equal to 26 dB.

The signals coming from the 4 receiving antennas are sent to the SP4T Hittite board EVAL HMC1084LC4 [26]. This board can select through an electronic control (control voltage ranging between 0 and −3 V), one of the four outputs that is sent to the BGT24MTR11 input (RFIN). The generated intermediate signals (IFI and IFQ) are acquired through the NI-USB 6361 DAQ card [27] and transferred to a PC to be used as input for the image reconstruction algorithms.

Various design solutions have been proposed in the literature for the radar radiating system, such as horn, Vivaldi, or dielectric antennas [28,29,30,31,32,33]. In this study, a simple radiating system based on a serial patch antenna array [34] was chosen to ensure high gain, light weight, and low cost (see Figure 4).

The realized serial patch array antenna is printed on a Roger RO4350B dielectric substrate (𝜀_𝑟_ = 3.66) with a thickness of 30 mils (762 μm), and a loss tangent of 0.0037. Since the array currents have to flow in the same direction, the feeding point is located as shown in Figure 4a. In this way, the microstrip line located below the feeding point shifts the current phase by 180° at the antenna central frequency, and therefore, on the two pairs of three patches, the current flows in the same direction. A taper on the patches’ width (along the radiating edge of each patch) was applied to reduce side lobe levels [35,36]. The taper factors are 0.75 and 0.5 for the last two patches located in the opposite directions. To take into account the parasitic effects introduced by the SMA connector-antenna transition, a fine-tuning using the Microwave Studio full-wave software by CST was performed. The optimized dimensions of the antenna microstrip lines are L_S_ = 3.5 mm and W_S_ = 0.5 mm. All the patches have the same length, L_P_ = 3.12 mm, while the widths are W_P1_ = 4.01 mm, W_P2_ = 3.0075 mm, and W_P3_ = 2.005 mm. The antenna radiation pattern, simulated with the full-wave CST software, is reported in Figure 4b and evidences a directivity of 14.2 dBi, with a side-lobe level of −14 dB. This value is 4 dB better than the one achieved simulating a uniform-width patch array and ensuring good rejection of environmental clutter.

The overall antenna system consists of a serial patch array antenna as transmitter (TX in Figure 5) and 4 serial array antennas, one wavelength spaced, as receiver (RX in Figure 5). The antenna was realized starting from the Gerber file generated by the CST software by using a numerically controlled milling table (Quick Circuit 5000).

The scattering parameters of the TX and RX antenna system, simulated by CST and measured using the Agilent PNA E8363C network analyzer, are reported in Figure 6. A good agreement between measurements and numerical results can be observed. In particular, the measurements show an impedance bandwidth (for a −10 dB threshold) of the transmitting antenna of about 600 MHz (between 23.8 GHz and 24.4 GHz) and a TX/RX coupling isolation greater than 35 dB.

The ADSimPLL software [37] was used to design the chirp shape and the loop filter components. The PLL parameters (maximum, minimum, reference and phase detector frequency, VCO tuning law, etc.) were considered to be the ADSimPLL inputs. In particular, since the PLL is controlled by a 16-time-scaled output frequency of the BGT24MTR11 board, the VCO central frequency is 24,125/16 = 1507.8 MHz, and the bandwidth 250/16 = 15.625 MHz. To reach a good compromise between the carrier phase noise and its settling time, a loop bandwidth of 45 kHz with a phase margin of 47° was chosen. To this end, a 3-pole 4th-order loop filter was designed using the ADSimPLL software. The filter schematic is reported in Figure 7a, while its experimental prototype with surface mount devices (SMDs) is shown in Figure 7b. The filter transfer impedance was analyzed by using Microwave Office software by Cadence. Figure 8 shows a comparison between the simulated results and the measurements performed on the filter prototype from which an average error of the order of 0.9 dB, corresponding to 11%, can be extrapolated. An excellent agreement between measured and simulated responses is evidenced.

To start up the radar, the registers of the ADF4159 PLL board are initialized by using the Analog Devices ADF4158-9 PLL software. With this tool, the radar chirp, identified with ADSimPLL software, was synthetized by means of 128 frequency steps, with a deviation of 122 kHz and a time duration of 8.12 μs. In this way, the time chirp duration is 1039 μs, and the total ramp frequency shift is 15.625 MHz.

To verify the actual operation of the PLL circuit used in the radar modular board, the ramp driving the VCO at the output of the loop filter was measured (see Figure 9), showing a chirp repetition frequency of 962 Hz, corresponding to a period of 1039 μs. As can be observed, the agreement between measurement and numerical results achieved from using the ADSimPLL software is good.

The phase noise on the tones synthesized by the PLL was estimated with the ADIsimPLL software. At the scaled output Q1 of the BGT24MTR11 block scheme shown in Figure 3, a Single-Sideband-to-Carrier Ratio (SSCR) of −105 dBc/Hz at 10 kHz offset frequency was obtained.

Finally, the NI-USB 6361 DAQ card was used to select the EVAL HMC1084LC4 switch outputs. A suitable voltage-level adapter is used to adapt the TTL output of the DAQ card to the inputs of the EVAL HMC1084LC4 board. All the operations of the NI-USB 6361 board are controlled by a PC on which a LabVIEW virtual instrument is running. The radar prototype, the used boards, the voltage adapter used to drive the SP4T switch, and the laser distance meter used as reference for the target distance are shown in Figure 10.

## 4. Numerical and Experimental Results

### 4.1. Test of the 2D-FT Algorithm on Canonical Cases

A first example of 2D-FT reconstruction test, based on the simulation of a point target located at *x_s_* = 0.5 m, *y_s_* = 3.0 m and with the ROI located between *x_min_* = −2 m, *x_max_* = 2 m, *y_min_* = 1 m, and *y_max_* = 5 m, employs an array of N_a_ antennas spaced by *d_a_* = *λ*_0_/2 (see Figure 11).

Under the hypothesis of point-like targets located far from the antenna array, the reflection coefficients $\Gamma_{pq}$ in (A17) take the following form:(14)$$\Gamma_{pq}\approx \sigma \;exp\left\{-j4\pi f_{q}\left(\frac{r_{p}}{c}\right)\right\}$$
where σ is a parameter linked to the target radar cross-section and to the electrical path of the RF signal, while $r_{p}$ is the target-to-antenna distance (see Figure 1).

To identify how the accuracy of the reconstructed target position depends on the fractional bandwidth and on the radar carrier frequency, three different cases were analyzed. In the first case, a radar operating at the central frequency of *f*_0_ = 24.125 GHz with a *B* = 250 MHz [*FBW* = 0.01 (1%)] was considered; in the second case, the radar exhibits a *B* = 2.4 GHz [*FBW* = 0.1 (10%)] at *f*_0_ = 24 GHz; and in the last case, the radar frequency is *f*_0_ = 2 GHz and *B* = 0.2 GHz [*FBW* = 0.1 (10%)]. Numerical simulations were performed by considering *N_a_* = 4 and *N_a_* = 16. The achieved error parameters, $E_{Q}$ and $E_{L}$, defined in Appendix A, are reported in Table 1 and Table 2, respectively.

From the above tables, it appears that a critical condition for the method (${E_{Q}}_{Max}$>> 1) is obtained with *f*_0_ = 2 GHz, *FBW* = 0.1, and *N_a_* = 16. To establish the suitability of these parameters in identifying the goodness of the method in finding the correct position of the target, the 2D-FT algorithm was applied, and the normalized intensity function versus position was reported in Figure 12a, along with the cross-range direction for the 4 antennas and 16 antenna cases. The figure outlines that, with 16 antennas, the 2D-FT technique is not able to identify the target position. In all other cases, the target is correctly identified. As an example, Figure 12b shows the cross-range cut of the normalized intensity function for the case *f*_0_ = 24 GHz, *FBW* = 0.1. In this case, both with 4 and with 16 antennas, the target position is correctly identified.

The tables also show that the condition based on the maximum average error in the entire ROI is less stringent and better suitable to characterize the performance of the 2D-FT method. Note that the quadratic term in (A2) is responsible for a worsening accuracy when the frequency is lower for the same *FBW*, while an increase in accuracy is observed when the antenna number decreases.

### 4.2. Target Identification with Simulated Data

The DAS, 2D-FT, and MUSIC algorithms were compared by considering the simulated scenario reported in Figure 13.

A point target (red dot) is placed 3 m far from the antenna plane. The radar, which is equipped with an array of *N_a_* = 4 antennas spaced at *d_a_* = *λ*_0_/2, exhibits a bandwidth of *B* = 250 MHz at 24.125 GHz.

Figure 14 shows the cuts of the normalized intensity function along the range (a) and cross-range (b) directions performed by using the three considered techniques.

The figures show that all the considered reconstruction techniques are able to correctly localize the target position with errors lower than 1 cm. In particular, the DAS and 2D-FT algorithms present a similar behavior, while the MUSIC, having super resolution properties, better identifies the target position (see Figure 14). For the considered radar, the theoretical resolution in range is *δ_r_* = *c*/2*B* = 60 cm, while the angular resolution is *δθ* = *λ*/(2∙*d_a_*∙(*N_a_* − 1)) = 0.33 rad (19°).

The half-power beam width (HPBW) in range and cross-range for DAS and 2D-FT is of 55 cm and 70 cm, respectively, which is very close to the 60 cm and 100 cm of the theoretical resolution (see Figure 14). In contrast with the two considered algorithms, the HPBW in range and cross-range of the MUSIC algorithm is of about 2 cm.

The considered algorithms required a computational time (CT) (MATLAB programs ran on a Pentium i7 processor) equal to the following: 2D-FT, 0.62 s; DAS, 2.94 s; and MUSIC, 14.5 s. Consequently, the 2D-FT algorithm using only two fast Fourier transforms has the best CT performance.

A two-target scenario consisting of two point scatters placed at a distance of 3 m from the antenna plane was considered to better highlight the resolution properties of the considered algorithms. The cross-range distance between the two targets was varied up to reaching, in *x* = 0, a −3dB value for DAS and 2D-FT algorithms (see Figure 15). This is achieved when the distance between the two targets is 1.2 m. This value is close to the theoretical cross-range resolution of the considered radar system (*δ_cr_* = 1 m at 3 m distance). Moreover, in this case, the super resolution properties of the MUSIC algorithm are also evidenced.

A further study was performed to evaluate the field of view (FOV) of the three considered techniques. To this end, the point target was moved along a line parallel to the antenna array. This means that different *θ* angles with respect to the antenna center identify the target (see Figure 13). Figure 16 shows the percentage error on the estimated distance with respect to the real one obtained for the three considered algorithms.

The figure shows that if the considered percentage error is 5%, the FOV is equal to ±16°, ±21°, and ±30° for MUSIC, 2FFT, and DAS algorithms, respectively.

Finally, it is interesting to note that the theoretical unambiguous angular section of the radar for $d_{a}=\frac{\lambda_{0}}{2}$ results in $\vartheta_{MAX}=\pm \frac{\lambda_{0}}{2d_{a}}=\pm 28.6{}^{\circ}$, which is very close to the simulated DAS FOV.

### 4.3. Target Identification with Measured Data

The realized modular radar was used to study a target location in a realistic environment. The considered scenario investigated consists of a 60 × 60 cm^2^ metallic panel placed at a distance (d) from the antenna plane (see Figure 17).

The signal coming from each antenna, sequentially selected with the help of an electronic switch (see Figure 3), is sent to the receiver, where a DAQ board, operating at a sampling frequency of 200 kHz, takes 208 samples over a chirp period of 1039 μs. Since the sawtooth signal used to generate the chirp introduces a consistent ripple level on the received signal, the first 28 samples are canceled. The same procedure is applied to the signals coming from the other three antennas, whose related data are stored in a 4 × 180 matrix.

Figure 18 and Figure 19 show the range and cross-range cuts of the intensity function, corresponding to targets placed 2.6 m and 5.4 m away from the antenna plane, respectively.

For the panel placed at d = 2.6 m, the achieved HPBW in range for DAS and 2D-FT was 55 cm (see Figure 18a), which is close to the theoretical resolution (60 cm), while for the HPBW in cross-range, values of 60 cm both for 2D-FT and for DAS were achieved. Both these latter values are close to the theoretical resolution of 0.33 × 260 = 85 cm. Finally, the MUSIC results outline a range and cross-range HPBW of about 10 cm.

The same result of 55 cm was obtained for the range HPBW when the metal panel was placed at d = 5.4 m, while values of 130 cm for 2D-FT and for DAS of cross-range HPBW were achieved, at which a theoretical resolution of 0.33 × 540 = 178 cm was related. In the same condition, the MUSIC algorithm gives rise to a range and cross-range HPBW of about 6 cm and 20 cm, respectively.

The achieved results for DAS and 2D-FT confirm that the range HPBW, being related to the signal bandwidth, does not change with the distance, while the cross-range HPBW, depending on the array–target distance, varies for the two considered cases.

To evaluate the effectiveness of the radar spatial discrimination, the experimental scenario consisting of two metal panels, as shown in Figure 20, was considered. Figure 21 shows the map of the intensity function achieved with the three considered reconstruction algorithms: DAS (a), 2D-FT (b), and MUSIC (c).

All the considered algorithms were able to identify the two metal targets with errors less than 20 cm. The hot spot produced by the closest panel in (−0.7 m, 2.8 m) is less intense due to the smaller panel size and lateral displacement, both of which are responsible for a lower back-scattered field toward the radar antennas with respect to the furthest panel. In particular, the radar cross-section of a square metal panel can be evaluated as follows [38]:(15)$$RCS=\frac{4\pi a^{2}}{\lambda^{2}}\left(\frac{sin\left(kasin\left(\theta \right)\right)}{kasin\left(\theta \right)}\right)$$
where “*a*” is the panel side, $k=\frac{2\pi }{\lambda }$ is the free-space wavenumber, and θ is the angle with respect to the panel normal. With reference to the square panels used in the experimental scenario reported in Figure 20, and taking into account the panel sizes and positions, the RCS seen by the radar is 19 m^2^ and 950 m^2^ for Panel 1 and Panel 2, respectively.

## 5. Conclusions

An FMCW radar, working at 24 GHz for short-range target imaging, realized by assembling the BGT24MTR11, ADF4159, and HMC1084LC4 Eval boards, and using a simple serial patch antenna array, was presented. A new, fast, and simple-to-implement 2D-FT algorithm was developed and tested on canonical scenarios. The proposed algorithm showed that a condition based on the maximum average error over the entire ROI is suitable to characterize its performance. The proposed algorithm was compared with the DAS and MUSIC algorithms. In particular, performances similar to those of the DAS were obtained, while lower resolution but better field of view were observed with respect to the results obtainable using MUSIC. Moreover, the 2D-FT algorithm is about 5 times faster than the DAS and more than 20 times faster than the MUSIC one. Another advantage of the radar proposed is that the number and the technology (printed, dielectric, and metallic) of antennas can be easily modified according to the design requirements. In conclusion the proposed radar system can be suitably employed for short-range real-time applications, such as the control of subjects and objects on the sidewalks of train and subway stations and the monitoring of elderly subjects in their home.

## Figures and Tables

**Figure 1 sensors-23-04119-f001:**
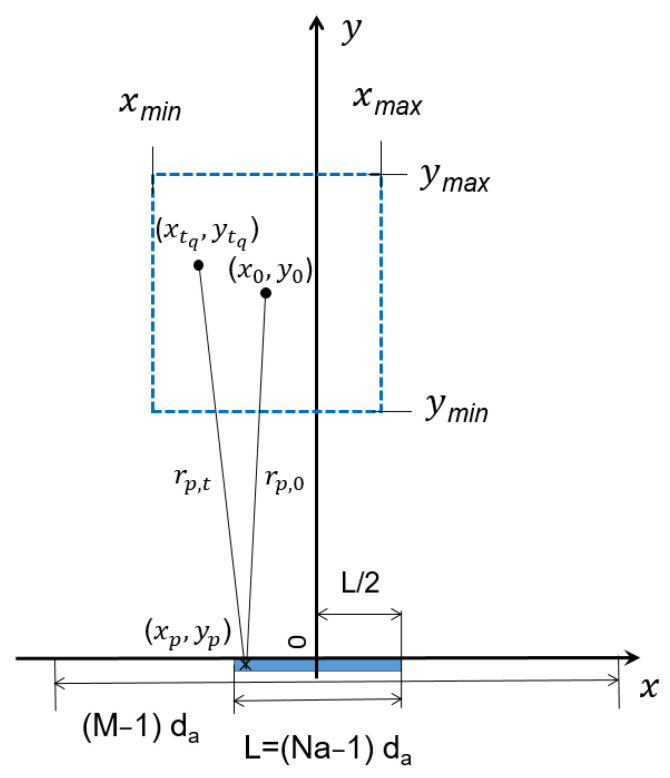
Considered geometry. Antenna array (blue region) and ROI (blue dashed line) are shown in the figure.

**Figure 2 sensors-23-04119-f002:**
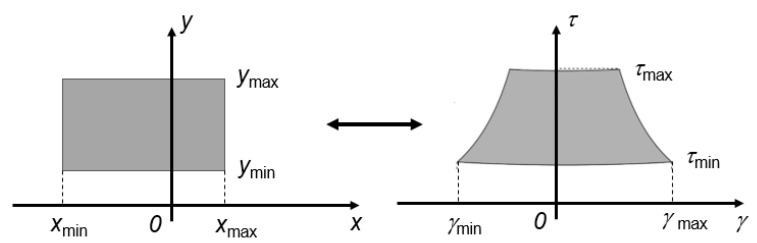
Mapping of the ROI in the corresponding region expressed by means of the γ and τ variables.

**Figure 3 sensors-23-04119-f003:**
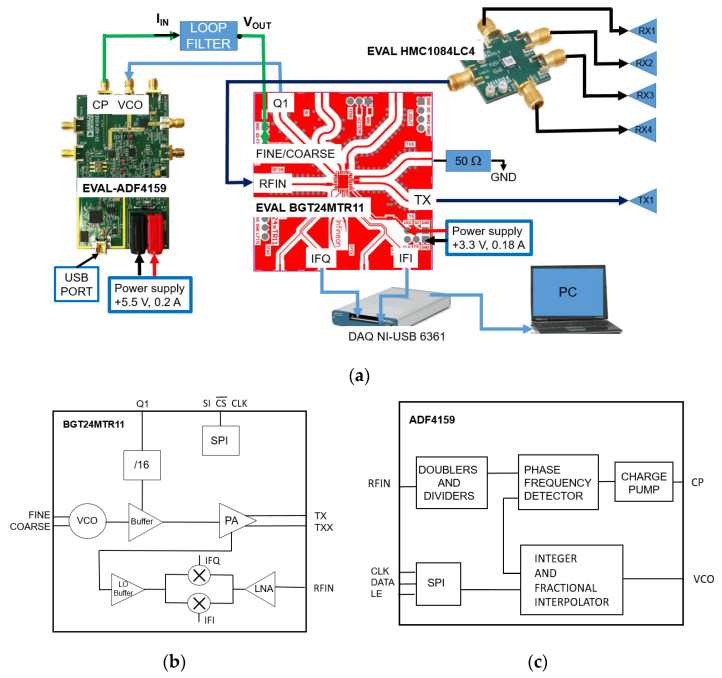
Block diagram of the realized radar (**a**). Block diagram of the BGT24MTR11 (**b**) and the ADF4159 (**c**).

**Figure 4 sensors-23-04119-f004:**
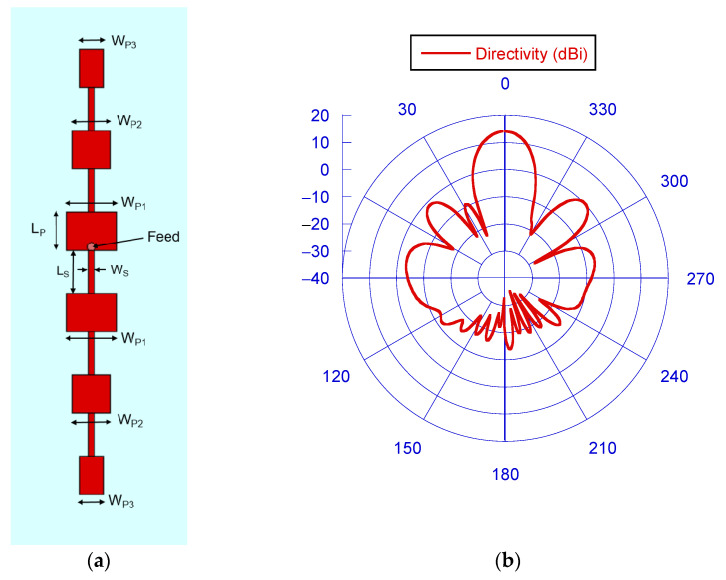
Schematic of the realized serial patch array antenna (**a**) and its radiation pattern (**b**). Theta = 0° corresponds to the normal at the air–dielectric interface of the array antenna.

**Figure 5 sensors-23-04119-f005:**
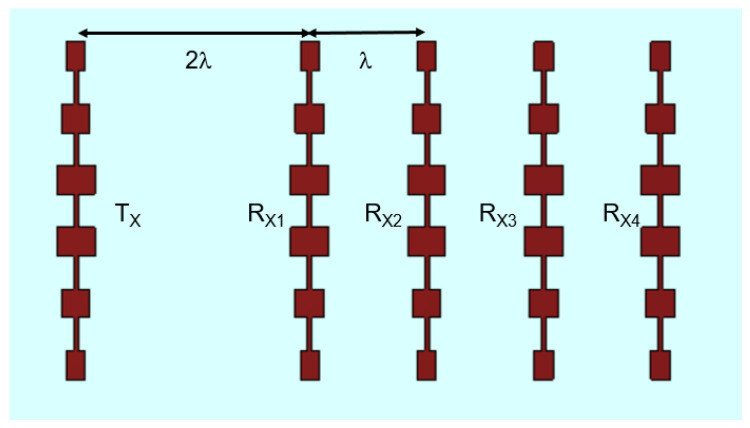
Schematic of the TX/RX antenna array.

**Figure 6 sensors-23-04119-f006:**
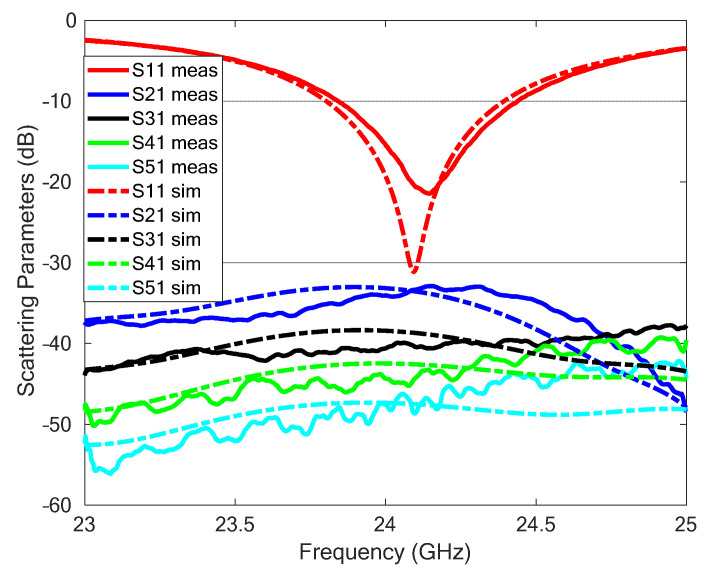
Measured (meas) and simulated (sim) scattering parameters of the array antenna system.

**Figure 7 sensors-23-04119-f007:**
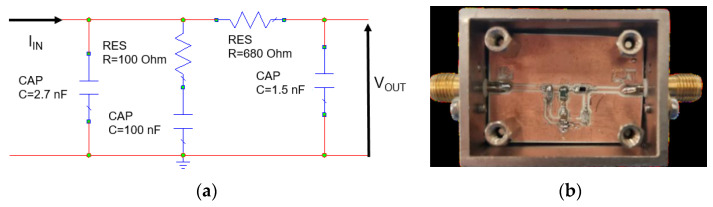
Schematic of the PLL loop (**a**) and its realization with SMD technology (**b**).

**Figure 8 sensors-23-04119-f008:**
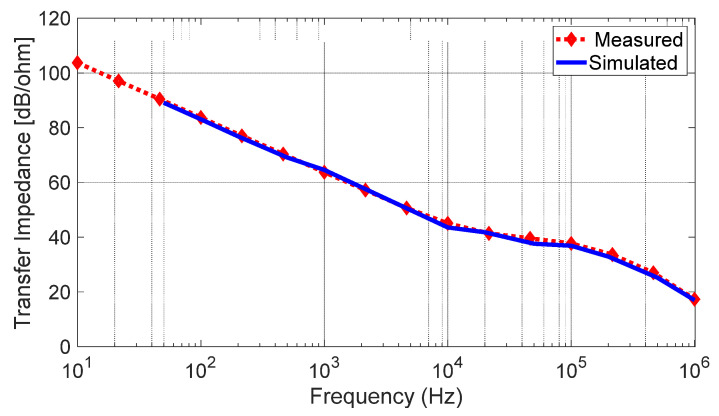
Measured and simulated responses of the loop filter.

**Figure 9 sensors-23-04119-f009:**
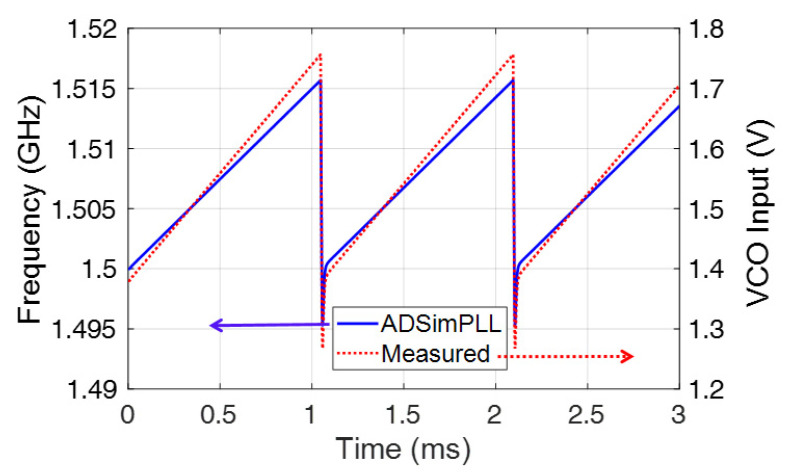
Measured ramp signal driving the VCO; and VCO output frequency as a function of time.

**Figure 10 sensors-23-04119-f010:**
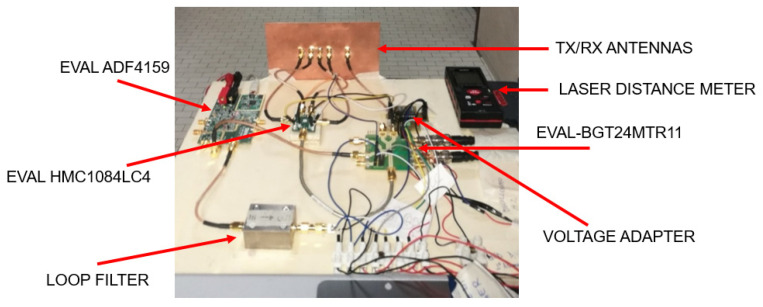
Prototype of the realized modular radar.

**Figure 11 sensors-23-04119-f011:**
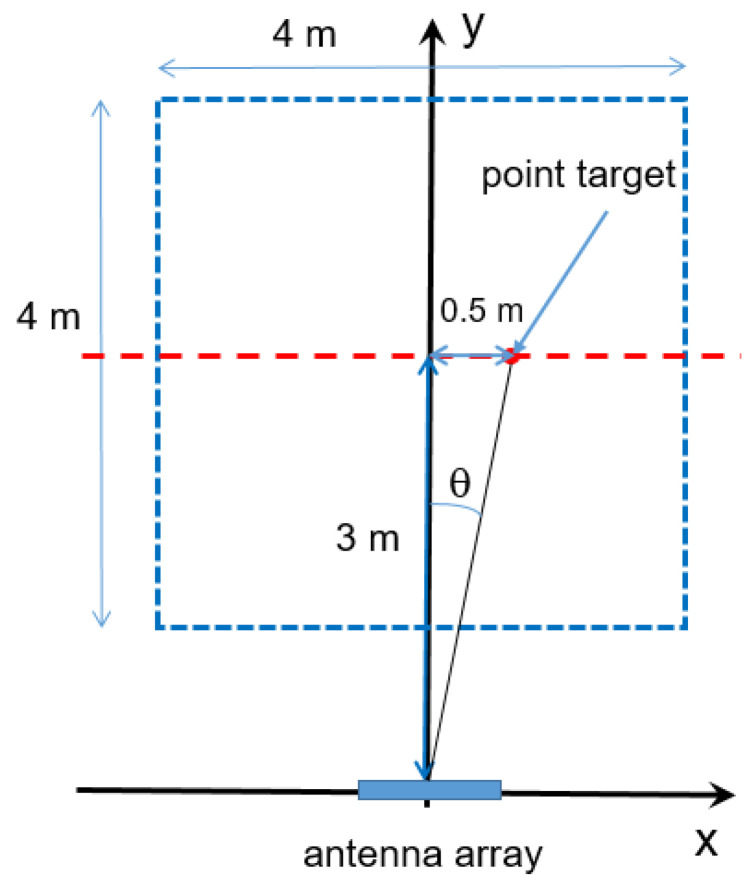
Geometry of the scenario: in the x–y plane. Point target and considered ROI (dashed blue line).

**Figure 12 sensors-23-04119-f012:**
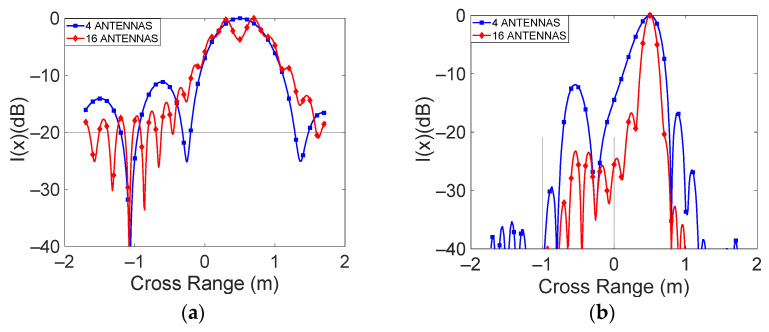
Cross-range cuts of the normalized intensity function for the 4 antennas and 16 antenna cases: (**a**) *f*_0_ = 2 GHz, *FBW* = 0.1; (**b**) *f*_0_ = 24 GHz, *FBW* = 0.1.

**Figure 13 sensors-23-04119-f013:**
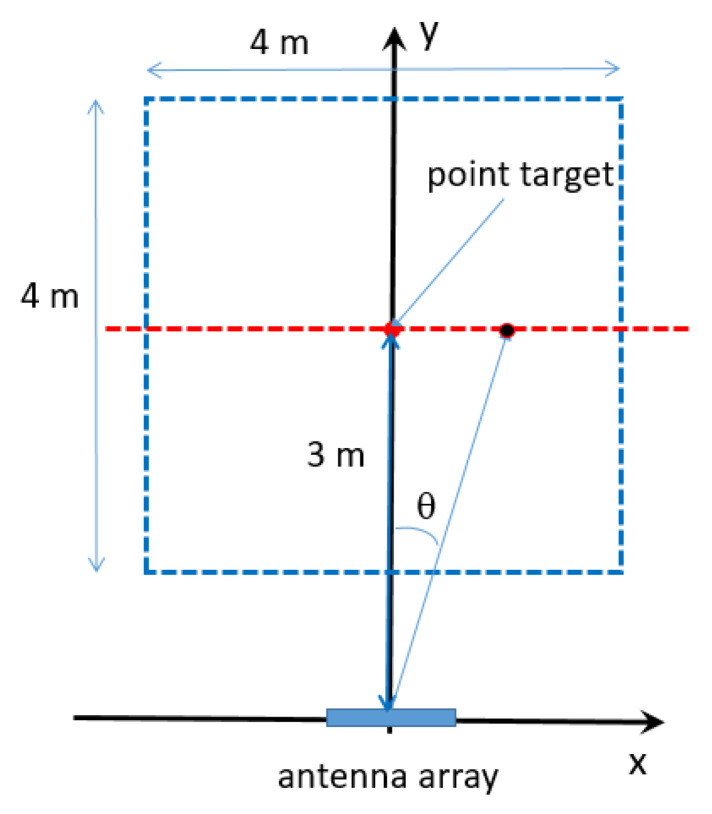
Geometry of the scenario in the x–y plane. Point target 3 m far from the antenna plane and considered ROI (dashed blue line).

**Figure 14 sensors-23-04119-f014:**
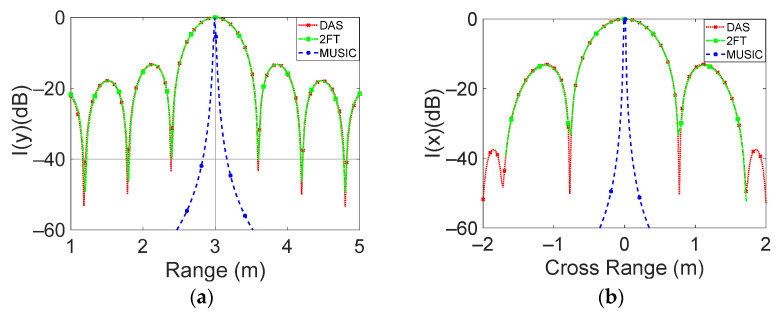
Comparison between the three considered techniques in the presence of a single target. The normalized intensity function versus (**a**) range and (**b**) cross-range distance is reported.

**Figure 15 sensors-23-04119-f015:**
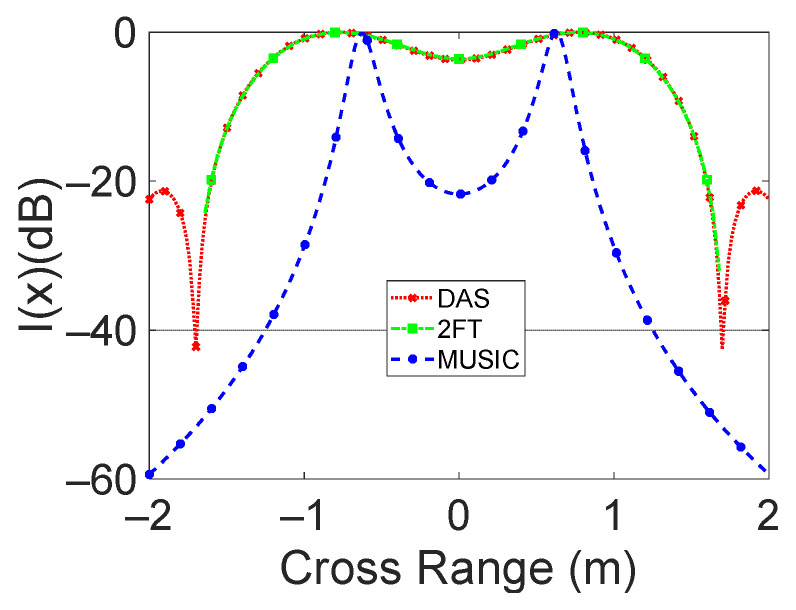
Comparison between the three considered techniques in the presence of two targets. The normalized intensity function versus cross-range distance is reported.

**Figure 16 sensors-23-04119-f016:**
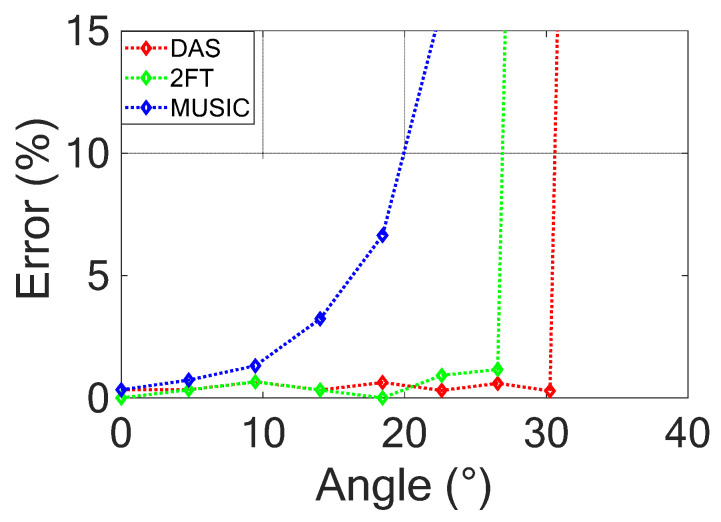
Target position error versus identification ϑ angle shown in Figure 13 for the three considered techniques.

**Figure 17 sensors-23-04119-f017:**
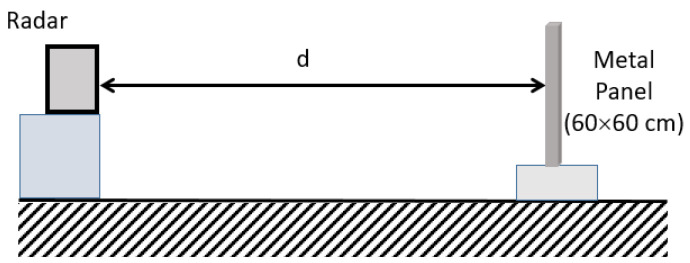
Sketch of the experimental investigated scenario.

**Figure 18 sensors-23-04119-f018:**
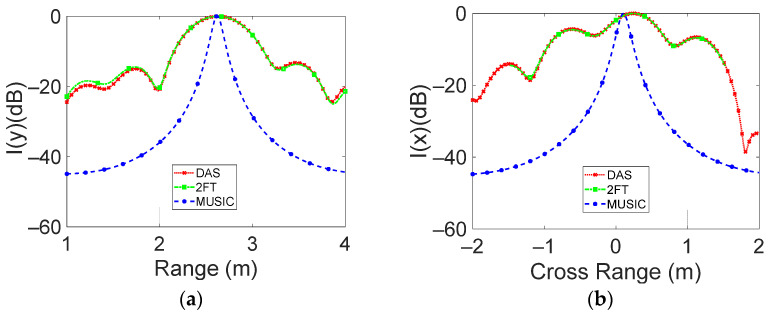
The normalized intensity function versus (**a**) range and (**b**) cross-range distance is reported. Panel distance is 2.6 m away from the radar antenna.

**Figure 19 sensors-23-04119-f019:**
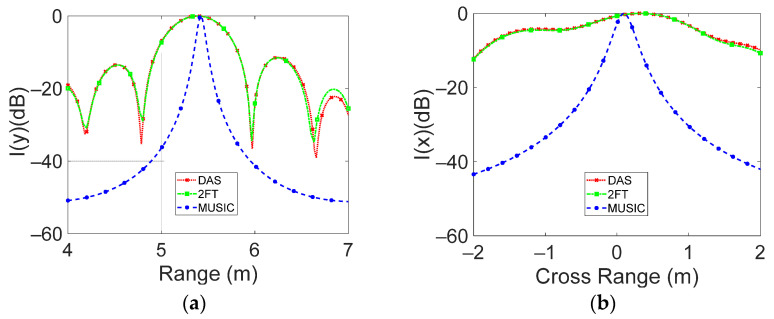
The normalized intensity function versus (**a**) range and (**b**) cross-range distance is reported. Panel distance is 5.4 m away from the radar antenna.

**Figure 20 sensors-23-04119-f020:**
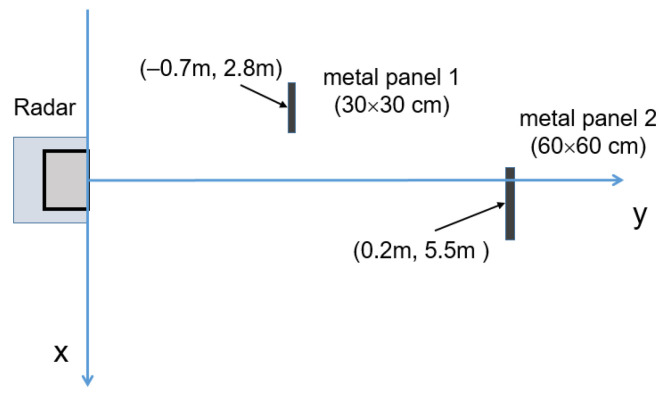
Sketch of the experimental investigated scenario with two square metal panel targets: metal panel 1 (30 × 30 cm) and metal panel 2 (60 × 60 cm).

**Figure 21 sensors-23-04119-f021:**
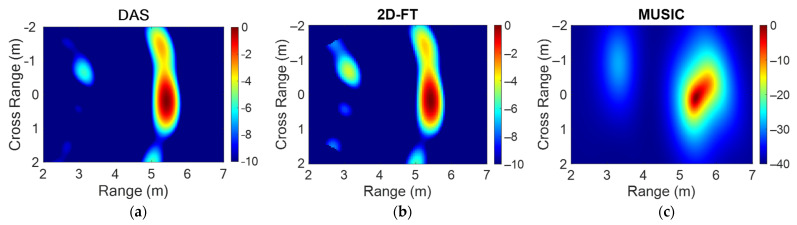
Map of the normalized intensity function in the ROI for the scenario in Figure 20 obtained with DAS (**a**), 2D-FT (**b**), and MUSIC (**c**) algorithms.

**Table 1 sensors-23-04119-t001:** Phase error produced by the approximation of the square term.

	${E_{Q}}_{Max}$	${E_{Q}}_{MeanMax}$
*f*_0_ = 24.125 GHz, *B* = 250 MHz *FBW* = 0.01, *N_a_* = 4	0.044	0.015
*f*_0_ = 24 GHz, *B* = 2.4 GHz *FBW* = 0.10, *N_a_* = 4	0.046	0.015
*f*_0_ = 2 GHz, *B* = 0.2 GHz *FBW* = 0.10, *N_a_* = 4	0.56	0.18
*f*_0_ = 24.125 GHz, *B* = 250 MHz *FBW* = 0.01, *N_a_* = 16	1.11	0.37
*f_0_* = 24 GHz, *B* = 2.4 GHz *FBW* = 0.10, *N_a_* = 16	1.16	0.37
*f*_0_ = 2 GHz, *B* = 0.2 GHz *FBW* = 0.10, *N_a_ =* 16	13.9	4.42

**Table 2 sensors-23-04119-t002:** Phase error produced by the approximation of the linear term.

	${E_{L}}_{Max}$	${E_{L}}_{MeanMax}$
*f*_0_ = 24.125 GHz, *B* = 250 MHz *FBW* = 0.01, *N_a_* = 4	0.044 $\left(\left.\frac{B}{f_{0}}\right|\begin{array}{l} =0.24\\ Max \end{array}\right)$	0.011 $\left(\left.\frac{B}{f_{0}}\right|\begin{array}{l} =0.95\\ Max \end{array}\right)$
*f*_0_ = 24 GHz, *B* = 2.4 GHz *FBW* = 0.10, *N_a_* = 4	0.42 $\left(\left.\frac{B}{f_{0}}\right|\begin{array}{l} =0.24\\ Max \end{array}\right)$	0.10 $\left(\left.\frac{B}{f_{0}}\right|\begin{array}{l} =0.95\\ Max \end{array}\right)$
*f*_0_ = 2 GHz, *B* = 0.2 GHz *FBW* = 0.10, *N_a_* = 4	0.42 $\left(\left.\frac{B}{f_{0}}\right|\begin{array}{l} =0.24\\ Max \end{array}\right)$	0.10 $\left(\left.\frac{B}{f_{0}}\right|\begin{array}{l} =0.95\\ Max \end{array}\right)$
*f*_0_ = 24.125 GHz, *B =* 250 MHz *FBW* = 0.01, *N_a_* = 16	0.22 $\left(\left.\frac{B}{f_{0}}\right|\begin{array}{l} =0.048\\ Max \end{array}\right)$	0.055 $\left(\left.\frac{B}{f_{0}}\right|\begin{array}{l} =0.19\\ Max \end{array}\right)$
*f*_0_ = 24 GHz, *B* = 2.4 GHz *FBW* = 0.10, *N_a_* = 16	2.10 $\left(\left.\frac{B}{f_{0}}\right|\begin{array}{l} =0.048\\ Max \end{array}\right)$	0.53 $\left(\left.\frac{B}{f_{0}}\right|\begin{array}{l} =0.19\\ Max \end{array}\right)$
*f*_0_ = 2 GHz, *B* = 0.2 GHz *FBW* = 0.10, *N_a_* = 16	2.11 $\left(\left.\frac{B}{f_{0}}\right|\begin{array}{l} =0.048\\ Max \end{array}\right)$	0.53 $\left(\left.\frac{B}{f_{0}}\right|\begin{array}{l} =0.19\\ Max \end{array}\right)$

## Data Availability

Data available upon request.

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
