# Peer review of "A Double Fourier-Transform Imaging Algorithm for a 24 GHz FMCW Short-Range Radar"

_sensors, 2023, doi:10.3390/s23084119_

Round 1
Reviewer 1 Report
1) No need to provide too much specifics at the abstract. Delete the details like 24 GHz multiple-input single-output frequency, 2FT, 20 times.
2) In Figure 2, what are the meanings of the blocks? Label them..
3) There is no need to show Figure 3b.
4) Equations 3 and 35 are a big unknown. Why do you want to show all details?.
5) Most derivations should be taken to APPENDIX.
Reviewer 2 Report
The authors submitted the paper titled, “A Double Fourier Transform Imaging Algorithm for a 24-GHz 2 FMCW Short-Range Radar” in Sensors journal.
In this paper, a 24 GHz multiple-input single-output frequency-modulated continuous-wave radar, for short range target imaging, assembling a transceiver, a PLL, a SP4T switch boards, and a serial patch antenna array, has been realized. A new algorithm, based on a double Fourier transform (2FT) has been developed, and compared with the Delay and Sum (DAS) and Multiple Signal Classification (MUSIC) algorithms available in the literature for target detection. The three reconstruction algorithms have been applied on simulated canonical cases evidencing radar resolutions close to the theoretical ones. The proposed 2FT algorithm is five time faster than DAS and 20 time faster than the MUSIC one.
Overall, the paper is well-written and organized. It can be accepted after addressing some queries. I have the following comments and suggestions for the improvement of the paper.
Can the authors provide the details of types of short-range radars based on double FT image processing? It is suggested to enrich the introduction by including recent studies on these radars (2020-2023).
The abstract is too generic. It is suggested to include some quantifiable technical data in the abstract. Other than comparing the speed of the proposed algorithm, what else can be compared? Please revise the context of the abstract.
Please check the polarity of the input current and the output voltage in Figure 5a.
How much (%) difference between measured and simulated transfer impedance (Figure 6)? Also, quantify the %difference regarding frequency.
It is suggested to centrally align all the equations for better representation.
Please eradicate typos and grammatical errors. For example, in the abstract, “applied on”, “five time” “20 time” etc. Such errors are consistent throughout the paper.
Reviewer 3 Report
Dear editor:
In this article, the author designs a new algorithm based on double Fourier transform to achieve fast imaging of 24GHz FMCW short-range radar. The algorithm proposed in the paper achieves amazing computational efficiency, which is an order of magnitude improvement compared to other algorithms. And the designed algorithm is verified by experiments. Although the work is solid, it should be worked on for readability, and I recommend accepting it with a major revision.
1 Regarding the numerical implementation of the proposed algorithm, I recommend that this part be placed in the appendix. The large formula affects readability, and the algorithm connotation has been shown in Section 3.1.
2: It is recommended that the font size and format in the picture be unified.
3: The core highlight of the article lies in the novelty of the algorithm. Please consider describing the content of the algorithm in advance, and then verify it on the designed radar system. Instead of describing the structure of the antenna system at the beginning.
Round 2
Reviewer 2 Report
I am satisfied.
Reviewer 3 Report
I recommend accepting it in its current format